# A Reverberation Suppression Method Based on the Joint Design of a PTFM Waveform and Receiver Filter

**DOI:** 10.3390/e24121707

**Published:** 2022-11-23

**Authors:** Lei Yue, Hong Liang, Tong Duan, Zezhou Dai

**Affiliations:** 1School of Marine Science and Technology, Northwestern Polytechnical University, Xi’an 710072, China; 2Kunming Shipbuilding Equipment Research and Test Center, Kunming 650051, China

**Keywords:** PTFM, reverberation suppression, estimate-before-detect, comb spectrum waveform cognitive filtering detection

## Abstract

Transmitting waveform design and signal processing method optimization are effective ways to improve a sonar system’s detection performance. In this study, the spectrum and ambiguity function characteristics of pulse trains of frequency modulation (PTFM) signals were deduced and analyzed to address the problem of serious reverberation interference in the detection of low-speed targets in shallow water environments. The action mechanisms of PTFM signal parameters on the comb spectrum and bed of nails ambiguity function were identified. PTFM signal parameters were designed according to reverberation suppression requirements. The threshold was calculated using the estimate-before-detect method, and the comb spectrum waveform cognitive filtering detection algorithm is proposed. The simulation and lake experimental results show that the PTFM signals’ reverberation suppression ability for low-speed targets was better than it was for stationary or high-speed targets. The proposed method has good universality, which can improve the output signal-to-reverberation ratio (SRR) by more than 6 dB.

## 1. Introduction

A sonar system’s performance is closely related to the characteristics of the environment and the target. If the transmit waveform and receiver can be optimized based on prior knowledge of the environment and target characteristics, interference can be significantly reduced, and target detection and recognition performance can be improved [1,2,3]. In a shallow water environment, reverberation is the main interference for active sonar; therefore, a sonar system’s emission waveform should have reverberation suppression capability [4]. A long continuous wave pulse and a broadband flat-spectrum pulse have good reverberation suppression abilities for high-speed and stationary targets, respectively [5,6]. Broadband comb spectrum signals are required to suppress low-speed target reverberation interference, including pulse train and frequency comb signals. Pulse train signals involve repeated periodic transmission of a kernel waveform at a pulse repetition interval, such as a PTFM signal. Frequency comb signals involve transmitting multiple tones, such as a sinusoidal frequency (SFM) signal, uniform comb (UC) signal, geometric comb (GC) signal, and Costas signal, simultaneously [4,7].

Broadband comb spectrum signals have the advantages of range–velocity joint high resolution, better suppression of the reverberation of low-speed targets [8,9], and reduction in Doppler sidelobes without losing Doppler resolution [10]. However, they have some disadvantages, such as distance-ambiguous sidelobes, and that the improvement of one signal performance reduces the other performance [8,9,10,11,12,13]. Prior to the design process, how the signal parameters affect detection capabilities such as reverberation suppression and target discrimination were studied. The ambiguity function was used to analyze the sonar signal’s reverberation suppression ability, distance resolution, and velocity resolution (defined as a two-dimensional correlation function for transmitting signals) [10,11,12,13].

Much research regarding frequency comb signals was conducted from the perspective of ambiguity function analysis. The ambiguity function characteristics of uniform and coprime comb spectrum signals were analyzed in [10]. Research presented in [11] focused on wideband Doppler-sensitive nonuniform frequency comb signals for sonar and exposed how bandwidth extent and tone count impact the resolution of velocity and distance. The ambiguity function characteristics of the SFM signal were analyzed, and its range ambiguity sidelobe suppression method was studied in [12,13]. Nevertheless, there is little research regarding the characteristics of pulse train signals, especially regarding how pulse train signal parameters result in spectral combing and the bed of nail ambiguity function. This situation is not conducive to the design of pulse train signal parameters and the optimization of signal processing methods.

Optimization of the comb spectrum signal processing method is essentially the design of the receiving filter. A comb spectrum signal correlation detection algorithm and a frequency–domain matching search algorithm were proposed based on the multi-peak structure of the comb spectrums in [14,15], which reduced the amount of computation, but could not estimate the target distance. These two methods did not thoroughly combine comb spectrum signal and reverberation characteristics; their detection abilities need to be improved. In addition, the methods of setting the threshold did not combine the target motion information in the traditional detection algorithm; this resulted in a decrease in detection probability when the comb spectrum signal was used to detect the moving target.

Given the above background, our motivation was the joint design of PTFM waveform and receiver filters to improve target detection ability in reverberation background, which could significantly improve the detection of low-speed targets in a shallow water environment. The main contributions of this research are as follows:We deduced and analyzed the PTFM signal’s spectrum, distance ambiguity function adjacent lobe spacing, ranging resolution, distance ambiguity function sidelobe level, and velocity ambiguity function sidelobe position.Based on derivation and analysis conclusions, we revealed the mechanism by which PTFM signal parameters affect the comb spectrum and nail plate ambiguity function. We then designed anti-reverberation PTFM signal parameters.To address the reduction in detection probability and further improve detection ability for moving targets that are problematic in the traditional threshold setting and signal processing methods, we introduced the estimate-before-detect threshold setting strategy and comb spectrum waveform cognitive filtering detection algorithm.

The remainder of this paper is organized as follows: In Section 2, we present our analyses of the PTFM signal spectrum and ambiguity function. The PTFM waveform and receiver filter were jointly designed and are presented in Section 3. In Section 4, the performance of the designed PTFM signal and the proposed detection method are evaluated, and the results are discussed. Finally, conclusions are presented in Section 5.

## 2. Analyses of the PTFM Signal Spectrum and Ambiguity Function

### 2.1. Derivation and Characteristic Analysis of the PTFM Signal’s Spectral Expression

The time domain expression of the PTFM signal is [16]
(1)$$s(t)=\sum_{n=0}^{N-1}\sin^{2}(\pi n/N)p(t-nT_{p})$$
(2)$$p(t)=\exp[2\pi i(f_{l}t+0.5B/T_{p}t^{2})],0<t<T_{p}$$
where Equation (2) is the sub-pulse of the PTFM signal, which is a linear frequency modulation (LFM) signal, $f_{l}$ is the start frequency, B is the bandwidth, $f_{0}=f_{l}+B/2$ is the central frequency, and $T_{p}$ is the sub-pulse duration. 

The Fourier transform corresponding to Equation (1) is
(3)S(f) =P(f)Y(f)
where P(f) is the Fourier transform of the sub-pulse LFM, and Y(f) is the Fourier transform of the envelope. 

Therefore, PTFM signal’s spectrum is comb-shaped, with a comb width of 4/T, and a comb spacing of $1/T_{p}$. The detailed derivation of the PTFM spectrum is provided in the Appendix A.

### 2.2. Derivation and Characteristic Analysis of the PTFM Signal’s Ambiguity Function 

The signal’s ambiguity function characterizes the radar or sonar signal’s ability to distinguish the distance and velocity of the target and to suppress clutter or reverberation. It is an effective tool for signal analysis and waveform design [16]. The broadband ambiguity function of the signal is expressed as [4,7]
(4)$$\Psi (\eta ,\tau )={\left|\sqrt{\eta }\int_{-\infty }^{+\infty }s(t)s^{*}[\eta (t-\tau )]dt\right|}^{2}$$
which can be realized in the frequency domain as [16]
(5)$$\sqrt{\eta }\int_{-\infty }^{+\infty }s(t)s^{*}[\eta (t-\tau )]dt=IFT[S(f)S^{*}(f/\eta )/\sqrt{\eta }]$$
where η=(c−v)/(c+v) is the Doppler compression factor, c is the speed of sound in the water, and v is the speed of the target. The distance ambiguity function Ψ(1,τ) is obtained when η=1. The velocity ambiguity function Ψ(η,0) is obtained when τ=0.

If we substitute Equation (1) into Equation (4), and we make η=1, then the distance ambiguity function of the PTFM signal is obtained as follows:(6)$$\begin{array}{l} \Psi (1,\tau )={\left|\int_{-\infty }^{+\infty }\sum_{n=0}^{N-1}A(n)p(t-nT_{p})\sum_{m=0}^{N-1}A(m)p^{*}(t-mT_{p}-\tau )dt\right|}^{2}\\ \;\;\;\;\;\;\;\;\;\;={\left|\sum_{n=m}^{N-1}W_{p_{n,}p_{m}}(1,\tau )+\sum_{n\neq m}^{N-1}W_{p_{n,}p_{m}}(1,(n-m)T_{p}+\tau )\right|}^{2} \end{array}$$

As the PTFM signal is periodic in the time domain, the interval between adjacent lobes of its range ambiguity function is $T_{p}$, and its ranging resolution is equal to that of the sub-pulse, namely, $\tau_{e}=0.44/B$.

The sidelobe level of the PTFM signal’s distance ambiguity function is only related to the number of sub-pulses, and the kth sidelobe level is
(7)$$\begin{array}{ll} SL_{k} & =10\mathrm{lg}(Sd_{k}/Sd_{0})\\ & =10\mathrm{lg}\{{\left[\sum_{n=1}^{N-1-k}\sin^{2}(\frac{n}{N}\pi )\sin^{2}(\frac{n+k}{N}\pi )\right]}^{2}/{\left[\sum_{n=1}^{N-1}\sin^{4}(\frac{n}{N}\pi )\right]}^{2}\} \end{array}$$
where *SL*_1_ = −2.2739 dB, *SL*_2_ = −9.5424 dB, and *SL*_3_ = −23.9225 dB; when N=5. When the time delay difference between multiple targets and the sonar platform is the sub-pulse width $T_{p}$, a false alarm and target distance estimation deviation easily occurs.

After deduction and simplification, the frequency domain expression of the PTFM signal’s ambiguity function was obtained as
(8)$$\begin{array}{l} {\left|S(f)S^{*}(f/\eta )/\sqrt{\eta }\right|}^{2}=\\ \;\;\;\;\;\;\;\;\;\;\;\frac{1}{16\eta }\;{\left[P(f)P^{*}(f/\eta )\right]}^{2}\left[X_{R}^{2}(f)+X_{I}^{2}(f)\right]\left[X_{R}^{2}(f/\eta )+X_{I}^{2}(f/\eta )\right]\; \end{array}$$

Figure 1 shows the mapping relationship between the comb spectrum signal spectrum and the pinboard ambiguity function. As $\left[X_{R}^{2}(f)+X_{I}^{2}(f)\right]$ is a sinc-like periodic function, when the Doppler compression factor η gradually increased or decreased from 1 during the process, the tooth peaks position of $\left[X_{R}^{2}(f/\eta )+X_{I}^{2}(f/\eta )\right]$ shifted, resulting in a periodic variation in which the overlapping area of $\left[X_{R}^{2}(f)+X_{I}^{2}(f)\right]$ and $\left[X_{R}^{2}(f/\eta )+X_{I}^{2}(f/\eta )\right]$ decreased first and then increased. This periodic change made the PTFM signal’s ambiguity appear to multi-peak along the Doppler compression factor axis. When the Doppler compression factor η changed so that the tooth peak position of $\left[X_{R}^{2}(f/\eta )+X_{I}^{2}(f/\eta )\right]$ shifted to the left or right $1/T_{p}$, the PTFM signal’s velocity ambiguity function had the largest side lobe peak. Therefore, the sidelobe position of its velocity ambiguity function could be calculated as
(9)$$v_{i}=\pm c/2T_{p}f_{0}.$$

## 3. Joint Design of the PTFM Waveform and Receiver Filter

### 3.1. Anti-Reverberation PTFM Signal Design

The design of the PTFM signal parameters includes pulse width, bandwidth, number of sub-pulses, etc. In this study, the optimal PTFM signal parameters were calculated using the constraint that the adjacent frequency points of the comb spectrum within a specific speed range did not produce aliasing, and that the maximum number of spectral lines could be simultaneously obtained using the following method:

When the maximum speed of the target’s movement is $v_{\max}$, to prevent aliasing of adjacent frequency points, it must be that
(10)$$2v_{\max}f_{0}/c<N/T.$$

Next, the number of sub-pulses can be taken as
(11)$$N=ceil[2v_{\max}f_{0}T/c].$$

Assuming that $v_{\max}=10\;m/s$, the PTFM signal’s pulse width is 25 ms, and its frequency modulation range is 10 kHz–20 kHz; then, N=5 is calculated using Equation (11). Figure 2 shows the designed PTFM signal. The PTFM signal comb spectrum had a comb tooth interval of 200 Hz and a comb tooth width of 160 Hz; its ambiguity function was in the shape of a bed of nails; its ranging resolution was 0.045 ms ≈ 0.44/B, *SL*_1_ = −2.2739 dB; *SL*_2_ = −9.5424 dB; *SL*_3_ = −23.9225 dB; and the maximum position of the velocity sidelobe was 9 m/s. It was consistent with the theoretical analysis presented in Section 2.2.

### 3.2. Comb Spectrum Waveform Cognitive Filtering Algorithm Design

According to the signal detection theory, active sonar target echo detection is a signal detection problem in which the transmitted signal is known, and the echo contains unknown parameters, including threshold setting and signal detection. Generally, the generalized likelihood ratio test, based on the maximum likelihood estimation principle, is used to complete threshold setting and signal detection, i.e.,
(12)$$\lambda_{G}(x)=\frac{p(x|{\overset{\wedge }{\theta }}_{1ml},H_{1})}{p(x|{\overset{\wedge }{\theta }}_{0ml},H_{0})}\begin{array}{c} \overset{H_{1}}{>}\\ \underset{H_{0}}{<} \end{array}\lambda_{0}$$
where $p(x|{\overset{\wedge }{\theta }}_{1ml},H_{1})$ and $p(x|{\overset{\wedge }{\theta }}_{0ml},H_{0})$ represent the probability density function of the observed data X when the parameter variables $\theta ={\left[\eta ,\tau \right]}^{T}$ reach maximum likelihood estimation with or without targets, respectively, and $\lambda_{0}$ represents the threshold.

Generally, when the maximum likelihood estimation (MLE) criterion is used to calculate the threshold $\lambda_{0}$, no target information is introduced and $\lambda_{0}$ is constant; this makes the detection probability of moving targets lower than that of stationary targets. Given this, the idea of track-before-detect [17] was used for reference in this study, and the estimate-before-detect method was used to calculate the threshold to achieve more reasonable detection of stationary and moving targets in the background of reverberation. That is, the target’s speed was informative, and the target’s distance was a nuisance.

Figure 3 shows the algorithm flow of comb spectrum waveform cognitive filtering detection, including echo processing and reverberation processing. Reverberation processing calculates the threshold, and echo processing detects the target and estimates the target’s speed and distance. The steps are as follows, and Steps 2 and 3 can be performed simultaneously.

**Step 1.** Comb filtering and target velocity estimation

A certain segment of data containing the target echo is taken from the autocorrelation and then subjected to Fourier transform. The target estimation speed $v_{n}$ is obtained through the comb filter bank $F={\left[F_{1},\ldots F_{m},\ldots F_{M}\right]}^{T}$ based on the MLE criterion.

**Step 2.** Test statistics acquisition and target distance estimation

Construct the frequency domain replica signal $S(\hat{\eta },f)$, and then complete the distance estimation and the test statistic acquisition of the target echo data according to the following equations:(13)$$X{\left(\tau \right)}_{mf}={\left|IFT\left[X(f)\cdot F_{n}\cdot S^{*}(\hat{\eta },f)\right]\right|}^{2}$$
(14)$$\hat{\tau }=\underset{\tau }{\arg\max}X{\left(\tau \right)}_{mf}$$
(15)$$H_{1}(k)=\max\left[X{\left(\tau \right)}_{mf}\right].$$

Select different segments of data containing target echoes to repeat Steps 1 and 2 to obtain the final test statistics containing target echoes.

**Step 3.** Calculate the threshold in the estimate-before-detect method

Calculate the Fourier transform of a specific segment of reverberation data after autocorrelation to obtain Y(f), multiply by the comb filter $F_{n}$ (this filter corresponds to the target’s estimated speed $v_{n}$), and use the following equation to calculate the test statistic in this segment of reverberation data:(16)$$H_{0}(k)=\max\left[{\left|IFT\left[Y(f)\cdot F_{n}\cdot S^{*}(\hat{\eta },f)\right]\right|}^{2}\right].$$

Select different segments of reverberation data to repeat this step to obtain the complete test statistics of the reverberation data and calculate the threshold $\lambda_{0}$.

**Step 4.** Threshold judgment and Pd acquisition

The final test statistic containing the target echo is compared with the threshold $\lambda_{0}$ to calculate the detection probability.

## 4. Simulation Results and Discussion

### 4.1. Computer Simulation Results and Discussion

Based on the point target echo and seabed unit scattering models, the scattering coefficients of seabed reverberation and target echoes at different speeds were numerically calculated. After convolving the seabed reverberation scattering coefficient with the transmitted signal, reverberation data were obtained. Changing the reverberation level could increase or decrease the SRR of target echoes.

Figure 4 shows the target echo, reverberation spectrum, and cross-ambiguity function at different speeds. As speed increased, its time domain waveform broadened; the low-speed target echo’s comb spectrum moved to the left by half a comb tooth interval, and the mid-speed target echo’s comb spectrum moved to the left by one comb tooth interval. The speed covered by the targets’ reverberation area ranged from −15 m/s to −7 m/s, −3 m/s to 3 m/s, and 7 m/s to 15 m/s.

As reverberation is formed by the superposition of scattered waves generated by a large number of random scatterers in the ocean to the incident acoustic signal at the receiving point, it is a stochastic process. To test and validate the joint design of the PTFM waveform and receiver filter from the perspective of statistics, we conducted experiments for each SRR in a Monte Carlo framework.

The number of Monte Carlo experiments was set to 10,000 times, with a false alarm probability (Pf) of 0.0001. Figure 5 shows the receiver operating characteristic (ROC) curve. When the detection probability was set to 0.9 and compared with the corresponding SRR value, it could be seen that the output SRR of the methods in [14,15], as well as the proposed method, were improved by 2 dB and 4 dB, respectively, at low speed (5 m/s) compared with mid-speed (9 m/s) and a static motion (0 m/s) state. This means that the designed PTFM signal had the best reverberation suppression performance in the low-speed motion state, which was consistent with the description of the velocity ambiguity function in Figure 2f. In addition, the detection performance of the proposed method was better than that of the methods in [14,15] at the target’s three relative motion states (0 m/s, 5 m/s, and 9 m/s); the output SRR increased by 6 dB, 7 dB, and 7 dB, respectively. Therefore, combined with the PTFM signal designed in this study, using the proposed method could obtain optimal detection results for low-speed relative motion.

### 4.2. Measured Data Results and Discussion

A computer simulation evaluated the PTFM signal design and detection method statistically, but it failed to simulate the amplitude–frequency response characteristics of the target echo. Therefore, the field experiment was designed to obtain the target echo in shallow water to further verify the effectiveness of the PTFM signal design and detection method. The experiment scheme is shown in Figure 6. The small target’s echo data and the lake bottom reverberation data were obtained, and the measured data were processed.

The hollow cylindrical steel tube was used as a small target for experimental detection; its echo frequency response characteristics were closely related to the acoustic scattering characteristics of the target. The acoustic scattering characteristics of the cylindrical target could be described by its target strength (*TS*), as shown in the following formula:(17)$$TS(f,\theta )=10\mathrm{lg}\left\{(\frac{aL^{2}}{2\lambda }){\left[\frac{\sin(kL\sin\theta )}{kL\sin\theta }\right]}^{2}\cos^{2}\theta \right\}$$
where a and L are the radius and length of the cylindrical target, respectively, *k* is the wavenumber, and θ is the angle between the incident sound wave and the normal direction of the target.

The spatial acoustic scattering characteristics of the cylindrical target are shown in Figure 7a, and the amplitude–frequency response of acoustic scattering at $\theta =0^{{}^{\circ}}$ is shown in Figure 7b. The acoustic scattering strength of the cylindrical target varied from the incident sound wave angle and reached a maximum at $\theta =0^{{}^{\circ}}$(the sound wave incident angle was 90°). The backscattering strength of the target increased with increasing frequency.

Figure 8 shows that the spectrum overlapped the target echo and reverberation. The target echo had a significant amplitude–frequency response in the 15 kHz~18 kHz band, and the reverberation had a significant amplitude–frequency response in the 10 kHz~15 kHz band. Compared with the stationary target, the comb-shaped spectrum of the low-speed target echo shifted to the left by half a comb tooth interval as a whole. The comb-shaped spectrum of the mid-speed target echo shifted to the left by a comb tooth interval as a whole, which was consistent with the previous analysis of the simulated target echoes.

The target echo-to-reverberation ratio (ERR) was calculated based on the algorithm in [14,15] and the proposed algorithm. The ERR logarithm was used to evaluate the parameter design of the anti-reverberation PTFM signal and the proposed method’s performance. As shown in Figure 9, when the input SRR was less than −6 dB, the detection output of the proposed algorithm was the largest, whether the target was stationary or moving. When the target speed was 5 m/s, the detection output SRR of the three methods reached the maximum, followed by the target speed of 9 m/s, and the output SRR was the smallest when the target was stationary. When the input SRR was −10 dB, the proposed algorithm’s detection ability improved by approximately 6 dB compared with the methods in [14,15].

Figure 10 shows the results of estimating the target distance using the comb spectrum waveform cognitive filtering algorithm. When the target speed was 5 m/s, the processing output sidelobe level was the lowest (lower than −20 dB). When the target speed was 0 m/s, the processing output sidelobe level was the highest, at approximately −9 dB. When the target speed was 9 m/s, the processing output sidelobe level (approximately −11 dB) was slightly lower than when the target speed was 0 m/s.

## 5. Conclusions

This study deduced and analyzed the spectral characteristics, ranging resolution, range ambiguity function sidelobe level, and velocity sidelobe position of the PTFM signal. It illustrated that the intrinsic relationship between the PTFM signal spectrum and ambiguity function is how the PTFM signal parameters affect the comb spectrum and ambiguity function characteristics. Based on theoretical analysis, PTFM signal parameters were designed according to the actual requirements of moving target detection in a shallow water environment. Next, the threshold was calculated using the estimate-before-detect method, and the PTFM waveform cognitive filtering detection algorithm was proposed. Finally, the computer simulation and field experimental processing results show the following:

(1) Against the background of shallow water reverberation, the designed PTFM signal had the best reverberation suppression ability for low-speed targets and could be combined with prior knowledge of the target’s motion to obtain design signal parameters with better reverberation suppression performance for targets within a specific speed range.

(2) Compared with traditional methods, the proposed comb spectrum waveform cognitive filtering detection method is universal for stationary and moving targets and can improve the output SRR by 6 dB or more.

(3) The proposed comb spectrum waveform cognitive filtering detection method could estimate the target distance. The anti-reverberation interference ability to estimate the distance of low-speed targets was better than that of stationary and mid-speed targets. The research in this paper has specific reference significance for other comb spectrum signal analyses and parameter designs.

## Figures and Tables

**Figure 1 entropy-24-01707-f001:**
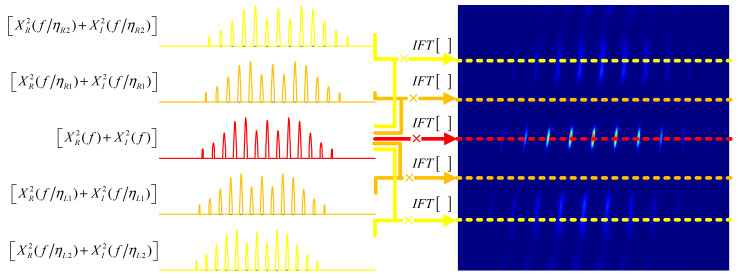
The mapping relationship between the comb spectrum signal spectrum and the bed of nails ambiguity function.

**Figure 2 entropy-24-01707-f002:**
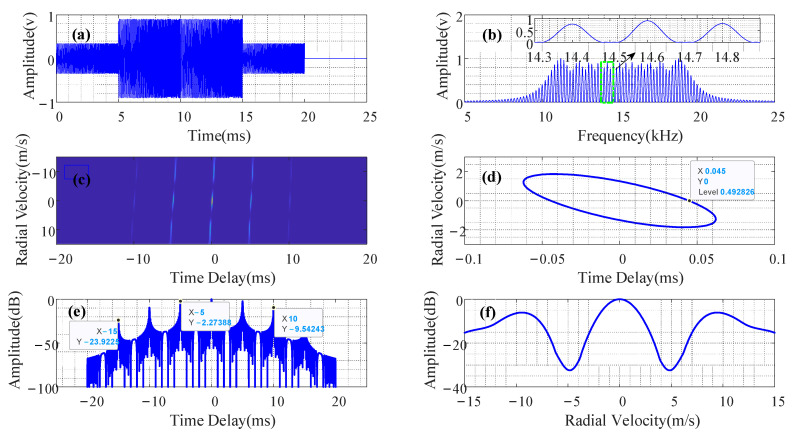
The designed PTFM signal. (**a**) PTFM in time domain. (**b**) PTFM in frequency domain. (**c**) Ambiguity function of PTFM signal. (**d**) Ambiguity function (−3 dB) of PTFM signal. (**e**) Time delay ambiguity function of PTFM signal. (**f**) Velocity ambiguity function (−3 dB) of PTFM signal.

**Figure 3 entropy-24-01707-f003:**
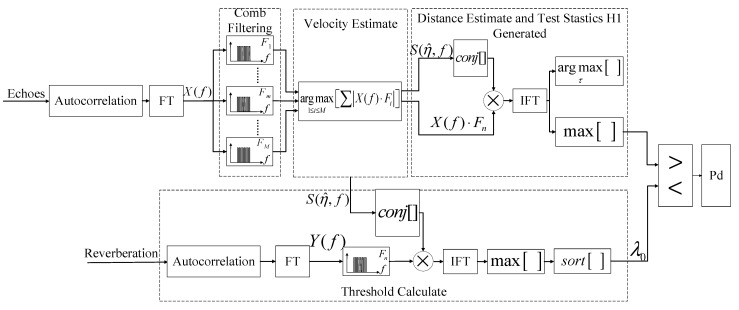
Comb spectrum waveform cognitive filter detection algorithm flow.

**Figure 4 entropy-24-01707-f004:**
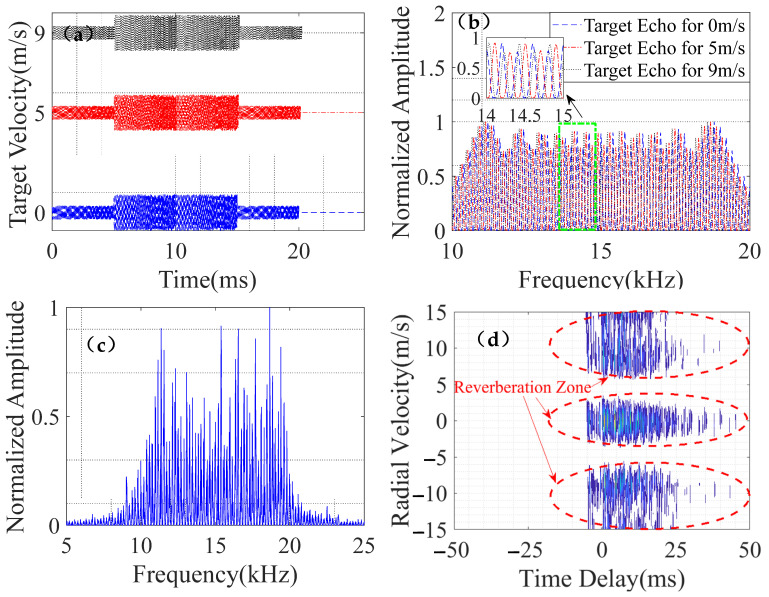
Simulated target echo and reverberation. (**a**) Target echo in time domain. (**b**) Target echo in frequency domain. (**c**) Reverberation in frequency domain. (**d**) Cross-ambiguity function of transmitted signal and reverberation.

**Figure 5 entropy-24-01707-f005:**
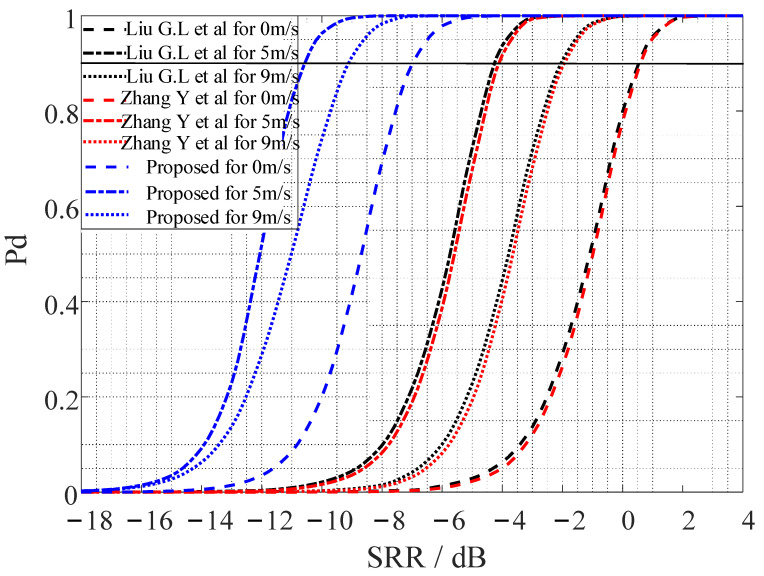
The Probability Target Detection versus SRR. Refs [14,15].

**Figure 6 entropy-24-01707-f006:**
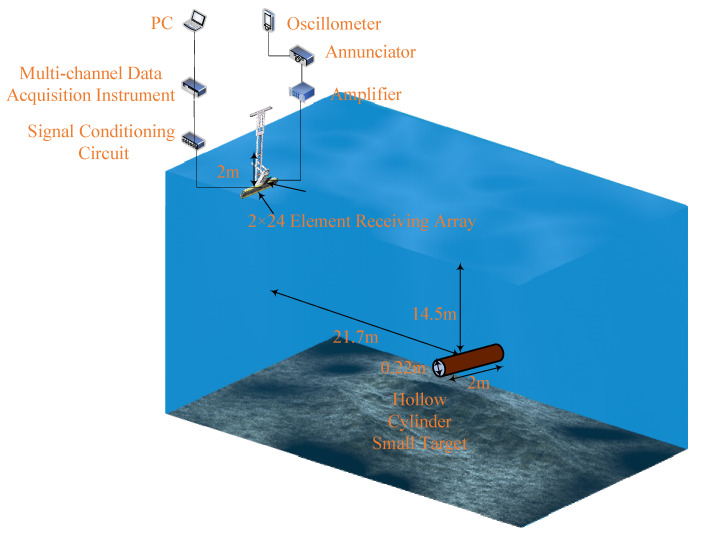
Experimental scheme for acquisition of target echo and lake bottom reverberation.

**Figure 7 entropy-24-01707-f007:**
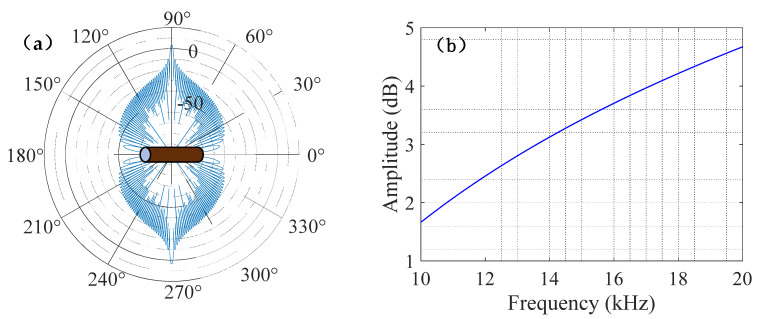
Acoustic scattering characteristics of the cylindrical target. (**a**) Spatial acoustic scattering characteristics of the cylindrical target. (**b**) Amplitude–frequency response of acoustic scattering at $\theta =0^{{}^{\circ}}$.

**Figure 8 entropy-24-01707-f008:**
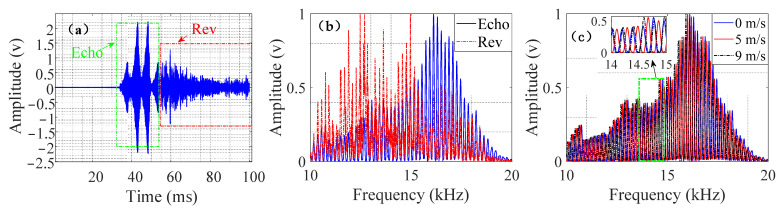
Measured target echo and reverberation. (**a**) Target echo and reverberation in time domain. (**b**) Target echo and reverberation spectrum. (**c**) Target echo spectrum at different speeds.

**Figure 9 entropy-24-01707-f009:**
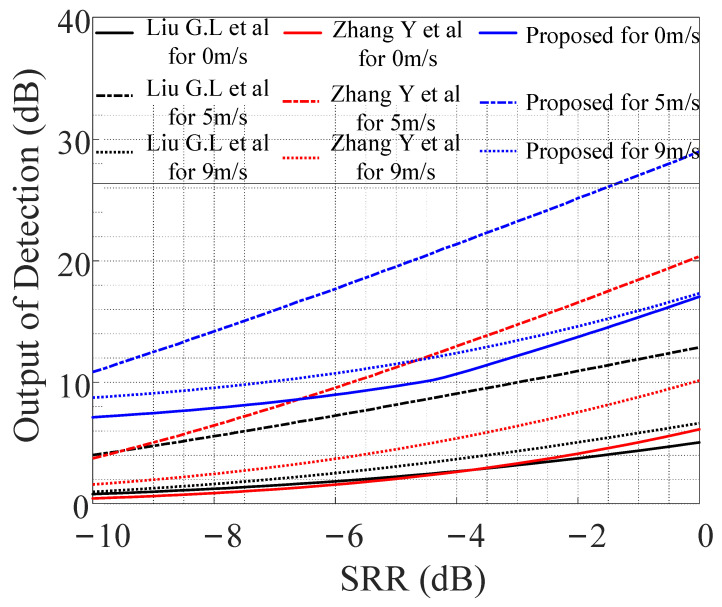
Detection output versus SRR. Refs [14,15].

**Figure 10 entropy-24-01707-f010:**
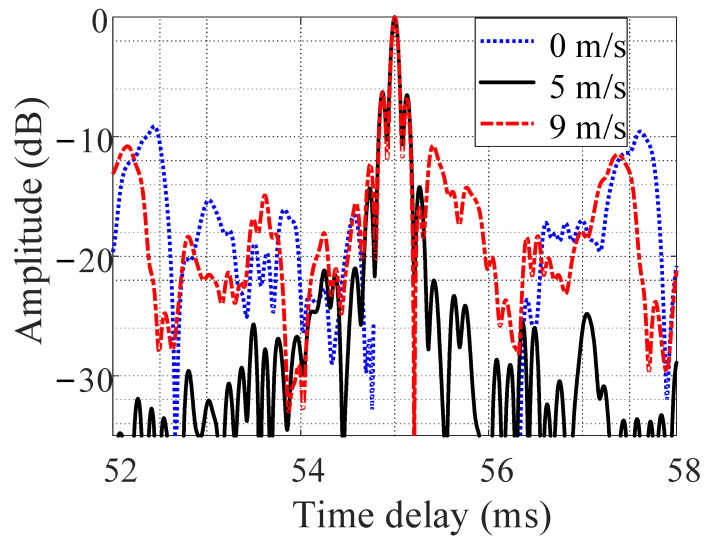
Target distance estimated using the comb spectrum waveform cognitive filtering algorithm.

## Data Availability

Not applicable.

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
