# Peer review of "A Reverberation Suppression Method Based on the Joint Design of a PTFM Waveform and Receiver Filter"

_entropy, 2022, doi:10.3390/e24121707_

Round 1
Reviewer 1 Report
Comments on “A Reverberation Suppression Method Based on the Joint De-2 sign of PTFM Waveform and Receiver Filter” submitted to Entropy-1965341
The paper in question can be potentially interesting after a very profound correction/improvement. In its actual form, the paper is not ready for publication.
After reading the manuscript, this reviewer has the following comments to the authors.
1. The original contribution of this study should clearly defined in the introduction.
2. The manuscript text abounds in jargon and very special terminology. It seems that it will be better to reduce the jargon and very special terminology in profit to a more general/mathematical/statistical terminology. The manuscript will be simpler to follow.
3. A typical detection problem with nuisance parameters is considered in the manuscript. Nevertheless, the authors do not explain which parameters are informative and which are nuisance (this Reviewer considers that the nuisance parameters are the distance to the target and the speed).
4. The authors use a Monte Carlo simulation to estimation the operating characteristics of the detector. Moreover, they speak about the probability of missed detection and the probability of false alarm. This means the existence somewhere of a stochastic model. This model is absent from the text. What is random? What is not random? Without such an explanation, the reading/understanding the results are impossible.
5. To continue the previous comment: the words “random”, “stochastic” are absent from the manuscript. It is impossible to interpret the obtained results without a precise description of a stochastic part of the model.
Author Response
Dear reviewers:
Thank you very much for taking the time to review this manuscript. We have carefully revised the manuscript ID: entropy-1965341 entitled “A Reverberation Suppression Method Based on the Joint Design of PTFM Waveform and Receiver Filter” and replied to the comments as follows. Thanks for your advice. All of your suggestions are very helpful for revising and improving our paper.
We have reviewed the manuscript according to your comments. Clearer statements and more accurate explanations have been given to improve the quality of the document further. Moreover, the responses to the reviewers are as follows. The corresponding changes in the manuscript are marked in yellow.
Comments 1:
The original contribution of this study should clearly defined in the introduction.
Response 1:
We combed the introduction again, enriched the research background, introduced this research according to the shortcomings of the research status quo, and wrote out the original contribution of this manuscript in the introduction.
Based on the above background, our motivation is the joint design of PTFM waveform and receiver filter to improve the target detection ability in reverberation background, and it can significantly improve the detection ability of low-speed targets in a shallow water environment. The main contributions of this research are:
- We deduce and analyze the spectrum, distance ambiguity function adjacent lobe spacing, ranging resolution, distance ambiguity function sidelobe level, and velocity ambiguity function sidelobe position of the PTFM signal.
- Based on the conclusion of derivation and analysis, we reveal the mechanism that PTFM signal parameters affect the comb spectrum and nail-plate ambiguity function. Then design anti-reverberation PTFM signal parameters.
- Aiming at the problem that the traditional threshold setting method and signal processing method reduces the detection probability, and further improves the detection ability for moving targets, we put forward the estimate-before-detect threshold setting strategy and comb spectrum waveform cognitive filtering detection algorithm.
Comments 2:
The manuscript text abounds in jargon and very special terminology. It seems that it will be better to reduce the jargon and very special terminology in profit to a more general/mathematical/statistical terminology. The manuscript will be simpler to follow.
Response 2:
We deleted the technical terms irrelevant to this study from the introduction and explained the technical terms involved to enhance the manuscript's readability.
Broadband comb spectrum signals are required to suppress the low-speed target reverberation interference, including pulse train and frequency comb signals. Pulse train signals involve repeated periodic transmission of a kernel waveform at some pulse repetition interval, such as PTFM signal. Frequency comb signals involve transmitting multiple tones simultaneously, such as (sinusoidal frequency) SFM signal, (uniform comb) UC signal, (geometric comb) GC signal, and Costas signal [4,7].
The ambiguity function is used to analyze the sonar signal’s ability of reverberation suppression, distance resolution, and velocity resolution, which is defined as a two-dimensional correlation function for transmitting signals [10-13].
Comments 3:
A typical detection problem with nuisance parameters is considered in the manuscript. Nevertheless, the authors do not explain which parameters are informative and which are nuisance (this Reviewer considers that the nuisance parameters are the distance to the target and the speed).
Response 3:
The detection in this study is a detection problem with parameter variables. The traditional detection threshold setting does not consider the target speed information and sets the threshold based on the maximum likelihood estimation criterion. Traditional detection threshold-setting methods have nothing to do with target speed and distance. This study adopts the estimated-before-detect strategy to set the threshold, and the target speed must be estimated first. For the detection in this study, the speed is practical information, but the distance is unnecessary.
According to the signal detection theory, active sonar target echo detection is a signal detection problem in which the transmitted signal is known, and the echo contains unknown parameters, including threshold setting and signal detection. Generally, the generalized likelihood ratio test based on maximum likelihood estimation principle is used to complete threshold setting and signal detection, i.e.
(12)
where and represents the probability density function of the observed data when the parameter variables reach maximum likelihood estimation with or without targets, represents threshold.
Generally, when the maximum likelihood estimation (MLE) criterion is used to calculate the threshold , no target information is introduced, and is constant, which makes the detection probability of moving targets lower than that of stationary targets. In view of this situation, the idea of track-before-detect [19] is used for reference, and the estimate-before-detect method is used to calculate the threshold to achieve more reasonable detection of stationary and moving targets in the background of reverberation. That is, the target's speed is informative, and the target's distance is a nuisance.
Comments 4:
The authors use a Monte Carlo simulation to estimation the operating characteristics of the detector. Moreover, they speak about the probability of missed detection and the probability of false alarm. This means the existence somewhere of a stochastic model. This model is absent from the text. What is random? What is not random? Without such an explanation, the reading/understanding the results are impossible.
Comments 5:
To continue the previous comment: the words “random”, “stochastic” are absent from the manuscript. It is impossible to interpret the obtained results without a precise description of a stochastic part of the model.
Response 4 and 5:
Added the description that reverberation is a random process. Added the description of the method. Such as the theoretical part of signal detection (Section 3.2), the Monte Carlo experimental method (Section 4.1), and the analysis of the different characteristics between computer simulation target echo and field experimental target echo (section 4.1 and section 4.2).
Based on the point target echo model and seabed unit scattering model, the scattering coefficients of seabed reverberation and target echoes at different speeds are obtained by numerical calculation. After convolving the seabed reverberation scattering coefficient with the transmitted signal, reverberation data is obtained. Changing the reverberation level can increase or decrease the SRR of target echoes.
Since reverberation is formed by the superposition of scattered waves generated by a large number of random scatterers in the ocean to the incident acoustic signal at the receiving point, it is a stochastic process. To test and validate the joint design of the PTFM waveform and receiver filter from the perspective of statistics, we conduct the experiments in a Monte Carlo framework for each SRR.
Computer simulation evaluated PTFM signal design and detection method statistically but failed to simulate the amplitude-frequency response characteristics of the target echo. Therefore, the field experiment is designed to obtain the target echo in shallow water so as to further verify the effectiveness of the PTFM signal design and detection method.
Yours sincerely
Lei Yue
Hong Liang
Tong Duan
Zezhou Dai
October 30, 2022

Reviewer 2 Report
The work looks interesting. I have two comments:
1. The equations are very dense. I would suggest for an easier reading to keep some important equations in the main text and move the rest to an appendix.
2. It wasn't clear to me how the simulation was carried out. The simulation results look quite different from the experiment. I assumed the simulation would be some sort of numerical modeling of the echo reflected from the immersed object, via, e.g. finite elements or finite difference in time domain. Clarification of this point could be helpful.
Author Response
Dear reviewers:
Thank you for your decision and constructive comments on my manuscript. We have carefully revised the manuscript ID: entropy-1965341 entitled “A Reverberation Suppression Method Based on the Joint Design of PTFM Waveform and Receiver Filter” and replied to the comments as follows. Thanks for your advice. All of your suggestions are very helpful for revising and improving our paper.
We have reviewed the manuscript according to your comments. Clearer statements and more accurate explanations have been given to improve the quality of the document further. Moreover, the responses to the reviewers are as follows. The corresponding changes in the manuscript are marked in yellow.
Comments 1:
The equations are very dense. I would suggest for an easier reading to keep some important equations in the main text and move the rest to an appendix.
Response 1:
We adjust the PTFM signal spectrum derivation (section2.1) and Derivation of the PTFM Ambiguity Function in frequency domain(section2.2) in the text to the appendix.
Appendix
Derivation of the PTFM spectrum
As shown in (3), the Fourier transform PTFM signal is
(18)
Where is the Fourier transform of the sub-pulse LFM, is the Fourier transform of the envelope, and
(19)
(20)
(21)
Equations (20) and (21) are sinc-like periodic functions, and is also sinc-like periodic function, which lead to spectral combing. Equation (20) is an even function, and Equation (21) is an odd function. Combined with the characteristics of the sinc function, it is not difficult to derive the period of is , the width of the main lobe and the grating lobe is , and the position of the tooth peak is ,.
Derivation of the PTFM Ambiguity Function in frequency domain
As shown in (18), the Fourier transform PTFM signal is
(22)
Replace by in (22), and take conjugation, then can be obtained
(23)
Multiply (22) and (23), and we can get
(24)
After taking the modulus square of (24), finally, we can get
(25)
Comments 2:
It wasn't clear to me how the simulation was carried out. The simulation results look quite different from the experiment. I assumed the simulation would be some sort of numerical modeling of the echo reflected from the immersed object, via, e.g. finite elements or finite difference in time domain. Clarification of this point could be helpful.
Response 2:
We added the description of the method. For example, as described in section4.1 about the simulation methods of target echo and reverberation, the differences between the evaluation standards of computer simulation and field experiment results are clearly stated.
Based on the point target echo model and seabed unit scattering model, the scattering coefficients of seabed reverberation and target echoes at different speeds are obtained by numerical calculation. After convolving the seabed reverberation scattering coefficient with the transmitted signal, reverberation data is obtained. Changing the reverberation level can increase or decrease the SRR of target echoes.
Yours sincerely
Lei Yue
Hong Liang
Tong Duan
Zezhou Dai
October 30, 2022

Round 2
Reviewer 1 Report
Comments on “A Reverberation Suppression Method Based on the Joint De-2 sign of PTFM Waveform and Receiver Filter” submitted to Entropy-1965341
The paper in question has been slightly improved after the first round of revision. It seems that the Authors understand the main concerns of this Reviewer. In this Reviewer’s opinion, the main problem of this manuscript now is English grammar and manuscript style. It is suggested a very extensive editing of English language and the style of manuscript before publication. After such an extensive editing many misunderstanding will be avoided.
Author Response
Dear reviewers:
Thank you very much for taking the time to review this manuscript. We have carefully revised the manuscript ID: entropy-1965341 entitled “A Reverberation Suppression Method Based on the Joint Design of PTFM Waveform and Receiver Filter” and replied to the comments as follows. Thanks for your advice. All of your suggestions are very helpful for revising and improving our paper.
We have reviewed the manuscript according to your comments. The responses to the reviewers are as follows.
Comments:
The paper in question has been slightly improved after the first round of revision. It seems that the Authors understand the main concerns of this Reviewer. In this Reviewer’s opinion, the main problem of this manuscript now is English grammar and manuscript style. It is suggested a very extensive editing of English language and the style of manuscript before publication. After such an extensive editing many misunderstanding will be avoided.
Response:
This manuscript has undergone English language editing by MDPI. The text has been checked for correct use of grammar and common technical terms, and edited to a level suitable for reporting research in a scholarly journal.
In addition, we replaced “&& Test Stastics” with “and Test Statistics” in the Figure 3, and repositioned Figure 10 underneath Figure 9, which make the manuscript style reasonable.
Yours sincerely
Lei Yue
Hong Liang
Tong Duan
Zezhou Dai
November 7, 2022
